# Dismantling colonial legacies: Decolonising research and teaching at the Health in Humanitarian Crises Centre, London school of hygiene and tropical medicine

Sali Hafez[1,2]*, Amber Clarke[2], Katharina Richter[3], Michelle Lokot[1], Althea-Maria Rivas[4], Neha S. Singh[1‡]

1 Health in Humanitarian Crises Centre, London School of Hygiene and Tropical Medicine, London, United Kingdom, 2 FAIR Network, London School of Hygiene and Tropical Medicine, London, United Kingdom, 3 School of Sociology, Politics and International Studies, University of Bristol, Bristol, United Kingdom, 4 School of Oriental and African Studies (SOAS), University of London, London, United Kingdom

‡ Principal investigator.
* Sali.hafez@lshtm.ac.uk

## Abstract

Despite a burgeoning discourse within the humanitarian health community regarding decolonisation, there remains lack of practical guidance for researchers seeking to decolonise their work. We conducted a qualitative study which aimed to explore the perceptions of research and teaching staff at the Health in Humanitarian Crisis Centre (HHCC) and their external partners —including humanitarian health researchers, practitioners, and donors— regarding how to decolonise research, teaching and partnerships at a leading global health Higher Education institution in the UK. We conducted 20 semi-structured interviews and 3 focus group discussions with HHCC members and external partners, including donors, academic institutions in conflict-affected and humanitarian settings, and practitioners from local and international humanitarian organisations. The first theme explored the concept of decolonisation itself, examining the disparate definitions and understandings held by HHCC members and partners, along with examining the institutional appetite and the role of leadership in driving decolonisation efforts. The second theme focused on sectoral and structural barriers to decolonising HHCC's work, including the dominance of Western-defined knowledge models, inequitable funding policies and practices, and epistemic injustice. Finally, the third theme explored HHCC's experiences in decolonising teaching and curriculum. The study identifies good practices within the HHCC community including knowledge co-production, equitable authorship arrangements, co-dissemination of findings, assigning co-principal investigators from conflict-affected countries, and centring and building on the experiences of researchers with relevant lived experience. However, these individual efforts contrast with a lack of appetite at the institutional level to address the underlying structural barriers. Our

**Data availability statement:** The data for this study represents transcripts of interviews which contain potentially identifying information. We are unable to share this data publicly because of restrictions by the Ethics Review Board and the consent process. Relevant, de-identified excerpts of the transcripts are included in the paper. Interested readers can request for data to the Research Ethics Committee at the London School of Hygiene and Tropical Medicine at Ethics@lshtm.ac.uk

**Funding:** This research was funded by Wellcome Trust Institutional Strategic Support Fund (ISSF) and the London School of Hygiene & Tropical Medicine (LSHTM) Research Culture EDI Grant 204928/Z/16/Z and funds provided by the LSHTM Executive Office through the FAIR x LSHTM commissioning process.

**Competing interests:** The authors have declared that no competing interests exist.

study provides the foundations for humanitarian health researchers and educators based in the Global North to begin to practically decolonise their work in the sphere of global/humanitarian health.

## Introduction

In recent years, the humanitarian sector has witnessed a growing discourse on decolonisation and the dismantling of colonial legacies embedded in its structures, practices, and narratives [1,2]. Within the humanitarian sector, humanitarian health researchers, educators, and practitioners are also drawing attention to how health-related humanitarian research and practice often reproduce unequal power relations—manifested through donor-driven agendas, the dominance of 'international' actors, and the marginalisation of local knowledge, resources and leadership [1,3]. Despite emerging efforts to promote equitable partnerships in humanitarian health research, teaching, and practice, progress remains uneven [3]. Aligning with Aloudat's [4] definition of decolonising humanitarian action, we prioritised centring the experiences of our research participants. This definition emphasises the creation of a humanitarian approach that dismantles Eurocentric ideologies, avoids perpetuating the oppression inherent in current power dynamics, and strives to move away from acting as an extension of colonial structures. We utilise the term "decolonising" in this study as a verb to signify the ongoing and ever-evolving nature of this process, as opposed to a static noun implying a fixed endpoint. We acknowledge that the terms 'Global North' and 'Global South' are imprecise and problematic [5]. This binary oversimplifies the diverse cultural, geographic, and historical contexts of different regions. Nevertheless, given the prevalence of this dichotomy in humanitarian, global health and academic discourses [6], and the fact that our participants used these terms, we do refer to them in this paper. We recognise that grouping countries with different race relations and colonial/imperial history is limited and inaccurate and can be perpetuate the "Foreign Gaze" [7]. In this paper, we additionally use "humanitarian settings" to describe a range of situations, including humanitarian crises, protracted emergencies, armed conflicts and refugee hosting contexts.

### Colonial legacies in the humanitarian sector

The persistence of colonial legacies within the humanitarian sector is evident in practices like "White Saviourism" [8]. White Saviourism is the perceived moral obligation for white volunteers, missionaries, and development or relief workers to lift and rescue non-white people in developing countries out of poverty and misery [9]. While this concept originates in colonial histories that constructed "whiteness" as a racialised identity tied to moral and civilisational superiority, it is increasingly recognised not as an expression of racial identity alone, but as a broader power-laden orientation [10]. This mindset—rooted in hierarchical relations of power and benevolence—can be adopted by anyone occupying a position of privilege or institutional authority, including local elites, who may also perpetuate saviour dynamics within humanitarian

responses [10]. Rather than examining or even addressing (their own roles in maintaining) structural inequality, racialised resource access and/or the capacity of racialised people to lead change themselves, White Saviourism ultimately normalises and naturalises White Paternalism [11,12]. For some, the White Saviour Complex "[demonstrates] the symbolic violence of racialised inequality" [13], while for others humanitarianism itself "acts as a salve for sustained racial discrimination and violence, working if not to entirely invisibilise racial hierarchies within suffering, then to make the racial underpinnings of such suffering acceptable through supposedly universal practices of care" [14].

Humanitarian practices continue to be structured on the White Gaze and a white saviour mentality [8,15]. The White Saviour Industrial Complex is comprised of a wide array of practices, policies, organisations, and institutions which reify structural and historical injustice by reasserting control over the majority world by "a complex system of predominantly white-led initiatives rooted in the preservation of dependence and powerlessness" [16]. In practice, these dynamics pervade the entire humanitarian sector. At the organisational structure level, a key inequity-driving practice is the limited participation of LMIC experts and community representatives in governance structures and advisory bodies of organisations focusing on improving LMIC health outcomes [17,18]. Language used across the humanitarian system reinforces discriminatory and racist perceptions of non-white populations [15]. For example, the term 'aid' itself infers that humanitarian action and funding is based on charity, without recognising colonial histories of exploitation. Research and programme design and funding also remain rooted in Western values and knowledge systems, which render resulting work a continuation of Western devised standards and leaves local knowledge devalued [15].

## Decolonisation efforts at the london school of hygiene & tropical medicine

Within and surrounding the London School of Hygiene and Tropical Medicine (LSHTM), a Global North-based institution with global influence, these debates are demonstrably present. LSHTM itself grapples with its colonial past, acknowledging its founding purpose as "an active agent in and conduit for the British Empire" [19], but those external and internal to the institution question whether these efforts go far enough [20]. Recognising this historical influence, some scholars and students at LSHTM have engaged in critical self-reflection, scrutinising their roles in perpetuating unequal power structures through research methodologies, funding mechanisms, institutional partnerships, and teaching practices. Particular sites of interrogation, challenge and reimagining include groups such as the Decolonising Global Health LSHTM (DGH-LSHTM) community, the Decolonising Global Health lecture series, the Decolonising the Curriculum Facilitators and Working Group and the Fight Against Institutional Racism (FAIR) Network, among others [21–23]. Despite these debates and efforts, a crucial gap exists – the lack of practical guidance for researchers seeking to decolonise their work. Existing literature offers minimal concrete steps or resources specifically tailored for researchers, particularly in humanitarian health [3].

A crucial step towards actualising decolonisation beyond the concept's mere use as a metaphor or aspirational goal [24] involves establishing concrete commitments and corresponding actions to be regularly discussed, and developing practices that actors can be held accountable to and that contribute to dismantle the structural inequalities embedded in institutions and this field.

To bridge this gap, the FAIR Network and Health in Humanitarian Crises Centre (HHCC) at LSHTM have collaborated to create a charter that sets out practical and specific commitments towards decolonising the Centre's research, partnerships and teaching, and established a framework to guide its implementation. The HHCC was founded with an aim of advancing health and health equity in crises-affected countries through research, education, and the translation of knowledge into policy and practice. The FAIR Network, a legacy of the 2020 Black Lives Matter protests, acts as an independent group dedicated to addressing racism and colonial legacies within LSHTM [25]. As part of developing the Charter and guidance, qualitative work was conducted to explore the perceptions of HHCC members and external partners and collaborators regarding a decolonial agenda and decolonial practice in humanitarian health, the historical and contemporary barriers to decolonisation in this setting, the facilitators for catalysing change, and any existing practices employed by

HHCC members in their work. This paper presents findings from our study and suggests how researchers and educators at Global North institutions in the field of global/humanitarian health can begin to practically decolonise their work.

## Gaps and divergent conceptions within the practical guidance on decolonising global/humanitarian health

Our previous scoping review synthesised the limited practical guidance available on decolonising global/humanitarian health research, partnerships, teaching, and organisational practices [3]. The review aligns with existing literature that reveals divergent understandings of what decolonisation entails. While decolonising practices can be broadly understood as efforts that highlight and decouple humanitarian practices from colonial power relations and logics [26], there remains a lack of clarity on what decolonising humanitarian aid and development might mean in practice. For some, "equity, diversity and inclusion (EDI)", "equitable partnerships" or "localisation" are constitutive of a decolonial agenda [27].

Discussions revolving around challenging power imbalances in humanitarian action gained prominence following the 2016 World Humanitarian Summit (WHS) Grand Bargain's commitments to "*making principled humanitarian action as local as possible and as international as necessary*" [28]. While widely lauded, the localisation agenda has faced criticism, with some expressing scepticism that it may not effectively challenge existing power dynamics but instead perpetuate them [29]. International humanitarian organisations still control funding and decision-making, limiting local actors to sub-contractual roles rather than equal partnerships, and reinforcing colonial legacies through externally defined standards and accountability frameworks [30,31]. Moreover, the influence of neoliberalism within the humanitarian system continues to pose a significant barrier to a more emancipatory and transformative agenda, and offers a predominantly technical solution to what is fundamentally a deeply political and contested issue of power and authority [31]. Overall, critics of localisation, and also EDI and equitable partnership efforts, highlight their distinct difference from decolonisation, notably the failure of some of these practices to address systemic barriers within the aid system, notably racism [2,29].

Our study includes participants who are from and based in both the Global North and the Global South, including those specifically in humanitarian settings, and provides further insights into how the practice of decolonisation is understood and experienced from different geographies and perspectives within the humanitarian health sector specifically.

## Methods

This study was conducted between 15/11/2022 and 24/03/2023 as part of a collaboration between FAIR and HHCC that aimed to demystify and operationalise the concepts of decolonial humanitarian research practices, teaching and partnerships at HHCC. The collaboration aimed to promote positive peer influence within the humanitarian aid and research community, foster a community of individuals committed to decolonisation and equitable practices in humanitarian aid and research, and encourage critical reflection on humanitarianism.

Three members of the FAIR Network led this study in consultation with the HHCC co-directors, ensuring an independent yet nuanced understanding of the HHCC's organisational context and working culture. The HHCC co-directors provided support through fund acquisition, support in designing research tools, and offered insights into the Centre's structure and members' work. Dr Althea-Maria Rivas joined the project as an expert consultant and provided her expertise in critical race, feminist and decolonial theoretical frameworks.

Alongside the scoping review [3], we conducted a qualitative study aimed at understanding how HHCC members and external stakeholders conceptualise and perceive the decolonisation of research, partnerships, and teaching. This qualitative inquiry sought to uncover the historical and contemporary barriers impeding efforts to challenge existing structures, while also identifying effective practices adopted by HHCC members [32]. The early findings of the scoping review contributed to informing the development of the interview guide —particularly the probing questions — and guided the initial analytical framework used to construct the codebook.

## Study design

The study team disseminated a call for participation through HHCC newsletters and LSHTM mailing lists. HHCC members who expressed interest were purposively sampled to ensure diverse perspectives across career stages and roles [33]. External stakeholders were similarly purposively selected from an existing list of partners, taking into consideration diversity in institutional affiliations, geographic locations, and positionalities relevant to HHCC's research, teaching, and partnerships.

We used semi-structured interviews and focus group discussions (FGDs) to explore the experiences of diverse stakeholders. A total of 20 semi-structured interviews encompassed 10 internal HHCC members, including senior, mid-career, and early-career academics, as well as research degree students and research project managers. Additionally, 10 external stakeholder interviews were conducted, capturing perspectives from donors/funders of humanitarian health research, partner academic institutions located within conflict-affected countries and humanitarian settings, and representatives of local and international humanitarian organisations. Participant characteristics are summarised in Table 1. All the participants self-defined their racial/ethnic background, occupation, grade of seniority and gender.

The study incorporated participatory elements, drawing on principles of participatory research to foreground the lived experiences and perspectives of HHCC members and partners [34]. While not fully co-produced, the research involved collaborative member engagement in FGDs, and co-design of the charter, thereby seeking to reduce extractive dynamics and centre community-defined knowledge [35]. Three FGDs facilitated an understanding of experiences, challenges, and practices among HHCC members. Four HHCC members participated in both the interviews and FGDs. The first FGD involved HHCC leadership, while the second included wider HHCC members. Subsequently, the third FGD took the form of a participatory charter design workshop to ensure member perspectives informed the development of the charter. This approach aimed to mitigate power imbalances within the research process while acknowledging the limitations of such methods in completely eradicating inherent power structures [36].

## Ethical considerations

The study information sheet was provided to all prospective participants, detailing the study's aim, expected contributions, potential risks, and benefits. All participants gave informed consent: written consent was obtained for all interviews, while verbal consent was secured for focus group discussions. Verbal consent was documented through the workshop recording. The data, including any quotations used in this article, was anonymised. The research team and participants

**Table 1. Study participants in the interviews.**

| Participants occupations | |
|---|---|
| Academic or humanitarian researcher | 12 |
| Humanitarian worker for International NGO | 3 |
| National or International donor agency | 3 |
| Humanitarian worker for local/national NGO | 1 |
| Other | 1 |
| **Total** | **20** |
| **Self-reported racial/ethnic backgrounds of participants** | |
| Arab | 2 |
| Asian | 2 |
| Black African | 7 |
| Mixed Asian | 1 |
| White | 8 |
| **Total** | **20** |

interacted in a collaborative manner, but no formal dependencies or hierarchical relationships influenced the data collection process. The London School of Hygiene & Tropical Medicine granted ethical approval for the primary qualitative data (reference: 28126/ 2022 on 11/11/2022).

## Data collection and analysis

The qualitative study was informed by participatory elements, prioritising the lived experiences and knowledge systems of participants, and seeking to critically examine how colonial histories and power structures shape the decolonisation of research, teaching, and partnerships [37]. Data were generated through interviews conducted online using Zoom, with 17 held in English and 3 in Arabic. All interviews were audio-recorded, and those in Arabic were translated into English to ensure consistency in the analysis. Following transcription by SH, the transcripts were inductively and deductively coded using MAXQDA software. Inductive coding involved a close reading of the data to identify patterns, meanings, and concepts emerging directly from participants' accounts. This process involved line-by-line coding, constant comparison, and iterative refinement of codes. Deductive coding was informed by sensitising concepts derived from the scoping review and the overarching research questions. Themes were subsequently developed through a process of synthesis and interpretation of the coded data by SH, with regular discussions among the research team to validate findings. While we intended to develop race-related themes, as suggested in the literature [3,15], participants ultimately did not focus extensively on this.

## Positionality of the research team

The co-authors of this research reflected on their power and positionality throughout this study. SH, ML, AR, and NSS identify as women of colour with personal and professional experience within humanitarian and development settings across various regions globally, and were informed by their own experiences of race, power and colonialism as practitioners. All the co-authors possess a deep understanding of colonialism's enduring effects on research practices and humanitarian action. While situated in an academic institution of the Global North, the co-authors strived to leverage their positionality to critically engage with these power structures. All the co-authors hold affiliations with British academia, and most of the co-authors are affiliated with LSHTM, an institution with a particularly overt colonial history. Finally, we acknowledge the power dynamics within the research team, where seniority often influences research hierarchies and decision-making processes.

# Results

This results section presents key themes related to decolonising humanitarian health research, partnerships, and teaching within the HHCC. The first theme explored the concept of decolonisation itself, examining the disparate definitions and understandings held by HHCC members and partners, along with examining the institutional appetite and the role of leadership in driving decolonisation efforts. The second theme focused on sectoral and structural barriers to decolonising HHCC's work. Finally, the third theme explored HHCC's experiences in decolonising teaching and curriculum. Within each theme, we highlighted the contrast between the perspectives of HHCC members and partners and the experiences of the study participants who identify as participants with lived experiences in humanitarian settings.

## Definitions, understanding and institutional readiness for decolonising the work of HHCC and LSHTM

**Definitions, understanding and meaning.** While there was a consensus on the need to dismantle colonial legacies within humanitarian research and practice, the understanding of the term "decolonisation" varied. Many HHCC members expressed uncertainty about whether the Centre or LSHTM had a definition of decolonising. One HHCC member suggested that the absence of a formal definition might not be coincidental. Another member shared: *"LSHTM has a definition for everything, and the absence or potentially hidden definition of what decolonisation means within our*

*institution doesn't seem like a normal process"*. When asked how they would describe decolonising humanitarian health research, participants mentioned several key dimensions: democratic decision-making processes, pragmatic research management and ownership, challenging entrenched power dynamics and structures, and addressing issues of racism and racial inequity. One HHCC member who identified as a person of colour explained: *"For me, decolonising our work means that programmes and research acknowledge that there is a power dynamic and then take measures to try to neutralise it"*. Some HHCC members advocated for co-ownership in research projects, while others viewed decolonising as a critical examination of power dynamics within partnerships. A junior academic of colour aptly underscored the distinction between defining decolonising humanitarian aid and research, as each confronts distinct yet interconnected structural barriers. Notably, only one senior white academic explicitly aligned decolonising with challenging existing power structures associated with racism and racial inequity.

Discussions within HHCC and LSHTM participants often conflated decolonisation with concepts like equitable partnerships, Equity, Diversity, and Inclusion (EDI), and localisation. One HHCC member described their understanding; *"[E]very institution now has an anti-racist action plan, an EDI policy. So, for the next couple of years, and beyond, all of these organisations are going to be able to say, "Oh, look, we're doing the work, let me show you our policy, let me show you our guideline, we just need more time to implement" … Some [organisations] will even call their work decolonial, but do they challenge power, at all?"*. Several participants perceived an institutional preference to reframe decolonisation using these alternative terms, potentially as a means of circumventing deeper engagement with the historical and ongoing power dynamics inherent in the concept. One HHCC member stated: **"There is still a very limited engagement with the underlying kind of politics, the underlying values, the underlying kind of moral foundations of anti-colonial action"**.

External HHCC partners from humanitarian settings defined decolonised humanitarian health research at the Centre as a value-based system focused on empowering not just their institutions but, ultimately, the communities they serve. One HHCC partner who identified as Black said: **"When we talk about decolonised humanitarianism, it [colonial humanitarianism] means an imbalance of power. It is a question of control. It is a question of voice. Whose voice matters? Whose voices are centred? Whose perspectives are centred? And, yeah, the extent to which there is really a locally grounded, community-driven community-centred approach to humanitarian health research"**. Some HHCC partners underscored the critical need to acknowledge how structural racism persists within the system, tracing its roots back to white saviourism that shaped humanitarianism and humanitarian health ecosystem. Donors, by contrast, leaned towards defining decolonisation in terms of equitable partnerships, decision-making and resource allocation. One donor emphasised the importance of dismantling structures built upon colonial objectives, noting: "*The entry point is around dismantling structures or institutions, which were built with the sort of focus on or the intention or the objective of that of this British colonial project, which is sort of taking resources or extracting for the purpose of benefiting global north or European institutions"*.

## Institutional readiness for decolonising research and partnerships

Most HHCC members perceived a limited institutional appetite for decolonising research practices and partnerships within the executive LSHTM leadership team. Despite recent debates at LSHTM and a stated commitment towards decolonisation following the publication of "LSHTM and Colonialism: A report on the Colonial History of the London School of Hygiene & Tropical Medicine (1899– c.1960)" [19], most HHCC members reported a disconnect between rhetoric and reality. One HHCC member said, "[The] very limited engagement with the work that [...] colleagues did on the history of colonialism in LSHTM was disappointing. And the report [Hirsch and Martin, 2022] itself was fantastic. The university's engagement with it has been disappointing. And I don't really see [...] inspiring, radical leadership within the institution when it comes to engaging with [...] what this means in practice". HHCC members attributed the limited interest in decolonising their work to the institution's capitalist business model, which is heavily reliant on research grants and prioritises preserving the status quo to benefit from this lucrative structure - as explained in the next section.

While leadership at LSHTM was stated to demonstrate limited enthusiasm for decolonisation efforts, HHCC members reported a more receptive environment within the Centre, with a greater willingness to engage with the decolonisation agenda. This openness manifested as a willingness to engage in critical discussions about decolonising research practices and their implications for research projects, methods and teaching. Some HHCC members credit this to the leadership of the centre co-directors, who are women of colour. One senior HHCC member said: "*The appetite and interest of the School in engaging in this discussion? not so much, but the centre may be, given the current management, they all seem to be interested and on board. But honestly, I don't know how much leverage the Centre on its own can have with such a contentious issue. I don't think the School is, is ready to push back with the big donors on something like that*". Other members further explained that while decolonising/decolonisation discussions are increasingly happening at the level of individual research projects and teams within HHCC, a more comprehensive centre-wide strategy and clear accountability mechanisms are needed to ensure consistent progress. Despite commitments, no study participants were aware of any existing accountability mechanisms for decolonising humanitarian health research, practice, partnerships, or teaching across other academic institutions, or humanitarian/global health organisations.

Partners from humanitarian settings often held a more positive view of the leadership and institutional appetite for decolonising humanitarian health research than some internal HHCC members. This optimism stems from their favourable experiences collaborating with specific HHCC research teams, which were likely individual initiatives rather than part of a broader institutional strategy. One HHCC humanitarian research partner based in Africa explained: "*I think that depends on your luck and the kind of the people you meet. If you meet the right people who have the right mindset, you have the opportunity to negotiate space and power. For this project (Name Anonymised), my collaborators and LSHTM were one of the best partners, because of this matter of power balance*". These positive experiences may have led external stakeholders to view HHCC as a more progressive entity compared to other research institutions, as explained by some donors. Some HHCC partners even viewed HHCC as an "*authentic ally*" in the decolonisation process, relying on HHCC to advocate for equitable partnerships with Global North funders and donors, given the leverage LSHTM and HHCC possess in these spaces.

### Structural barriers to decolonising research and partnerships

Participants discussed how efforts to decolonise humanitarian health research and teaching at HHCC are restrained by the very power structures it seeks to dismantle. The historical stain of "White Saviourism" and missionary work continues to impact humanitarian aid and research. One HHCC partner (donor) explained: "*What I noted was that there was very much this centring of this white saviour model in humanitarian research since I started working in this field 20 years ago… but I think that that's not changed that much for the centre or other institutions. But there's sort of that idea of not really questioning the framework where they work*". A closer look reveals a web of colonial power imbalances woven into the fabric of academia and research cycle, as one HHCC partner from a humanitarian-affected country explained: "*The … colonial history of LSHTM, and how the school benefited from the colonial resources from colonised countries and funding shape much of LSHTM's current structures, systems and norms... that's why the HHCC, being by nature delivering studies in humanitarian settings, or not even working in the UK, is by nature has the likelihood to employ some of the colonial approaches*".

Participants shared how these structural barriers were manifested across the research cycle, primarily benefiting HHCC/LSHTM, as outlined in the following sub-sections. These barriers perpetuate a cycle of knowledge hoarding, as the flawed colonial model fixated on serving institutional self-interest, fostering academic careers, and publications discouraging the transfer of valuable knowledge to the institutions and communities most impacted by research. Despite the intent to focus discussions on race, only two HHCC members identified race and racism as a structural barrier to decolonising their work. Few external partners identified race, racism, anti-blackness and whiteness as structural barriers that are particularly relevant for the humanitarian ecosystem.

## Priorities, research questions and design

HHCC members and partners frequently discussed a HHCC practice that reflects colonial power dynamics, specifically, the tendency for researchers to dominate the research agenda, setting priorities and determining study questions with limited engagement from partner institutions and minimal involvement of target communities. According to some HHCC members, decision-making regarding research viability often falls to HHCC/LSHTM. One HHCC member elaborated: *"Thinking of just decision making, who decides that the research is a viable option to submit for funding, I think that has come down -in my experience- to us at LSHTM, who have decided. That we are the researchers, so we have more experience in deciding what research needs to be done".*

While few HHCC members cited priorities derived from their lived experiences or past collaborations, some partners highlighted the lag between current practices and aspired locally-led research prioritisation, especially in complex emergencies. One HHCC member described the research priority setting process as: *"All designed at Keppel Street [LSHTM address]"* while a HHCC partner described their perspective on community-led priority setting: *"The ideal partnership and what I would hope for is when these communities define the priority areas of work or research. They're the ones to identify the real problems they face. And then maybe, throughout the process, maybe the institutions like HHCC or donors, then they can get involved but the initial thoughts have to come from these communities. Now, we are not there and not even close".*

Despite the challenges in decolonising research practices, several key strengths and good practices emerged. Many participants mentioned that upon securing grant funding, co-planning and co-designing activities are conducted collaboratively with partners during the study inception and planning phases. The co-planning/co-designing involves contextualising the study questions, research design, methods, site selection, clear role definition, decision-making, governance structure, and/or research management. One HHCC partner described the process: *"Imagine when we met in (Place anonymised) for our planning meeting, that was the first meeting. The first things we discussed were decision-making and decolonising the data. And some of the attendees were saying what is decolonising? Some of them even, they never heard of it. We don't talk about that as doctors".* Such collaborative practices foster a sense of ownership among partners and local communities, making space for their rightful role in meaningfully influencing the research agenda. Additionally, two HHCC members highlighted the value of employing shared leadership and co-principal investigators (co-PIs) for research projects. This structure, particularly when one co-PI is affiliated with a local institution in a humanitarian setting, facilitates shared decision-making and promotes a more equitable distribution of power within the research team.

External members from humanitarian settings particularly commended the use of steering groups within research projects. These steering groups serve as a governance structure for the research study, ensuring equal representation from each participating institution and facilitating high-level strategic decision-making, as described by participants. The steering groups incorporate power-sharing mechanisms, with leadership roles rotating among the partner institutions. This structure fosters shared decision-making, enhances leadership, and promotes project ownership. Moreover, the steering groups facilitate transparent communication and uphold good governance throughout the research lifecycle. One HHCC partner explained: *""When it comes to [allocating the grant and resources and] discussing key issues, [...] power [is] always shared with other members of the steering committee. This gave us confidence that we are working in [an] equal manner. [...] The seat of the steering committee rotates every year. That, to me, is one aspect that LSHTM project was one of the best, because of this matter of power balance".*

## Funding

There was consensus that funding requirements and donor conditionality represented significant historical and current structural barriers to decolonising research at HHCC and LSHTM. Most HHCC members and external stakeholders explained how the market-driven nature of Western academia and the relentless competition for funding incentivises a partnership model that benefits HHCC and LSHTM at the expense of the institutions working in humanitarian settings. Funding structures often channelled resources away from local institutions. One HHCC academic partner said: "British

and European funding was established in a way that benefits LSHTM, and of course HHCC. Local institutions don't have direct access to the donors themselves. Most donors [...] don't want to fund local institutions directly. They want to fund international institutions, and then the fund was streamed down to the local institution [...]. Cost-wise, it is not efficient. And it's not as effective either".

Some HHCC members and partners acknowledged how HHCC benefited from the existing funding structures, previous relationships with donors/funders, and Euro-centric grant requirements and conditions. Study participants from humanitarian settings explained how the short grant application cycles made navigating long applications and building partnerships nearly impossible. Participants mention that major UK donors required a significant research grant portion to be spent within UK institutions, as explained by HHCC members. Moreover, one HHCC member further elaborated on the financial resource allocation among research project consortium partners: *"The decision-making around funding from my experience has been pretty unilateral. I think that's the right word. It's been LSHTM teams, taking the decisions on funding allocations"*.

Participants explained that risk management practices and due diligence procedures created significant entry barriers for smaller, local, and grassroots organisations. HHCC partners described this as the *"instrumentalisation of due diligence structures"* to benefit specific institutions. A few HHCC members explained how due diligence requirements often portray support for non-Western institutions as inherently riskier, reinforcing a problematic colonial narrative. One HHCC member explained: *"There's a benchmark of less trust in non-Western institutions. I work with (African Partner) in multiple projects for the last 5 years, and every single time we have a project awarded, it feels like they have to go through due diligence all over again, when it's like we always work with them. They were never flagged for fraud or misconduct. This is not the same for European institutions"*.

The majority of participants, both HHCC members and partners, emphasised that donors bear responsibility for driving the decolonisation of humanitarian health research. Donors highlighted promising examples, such as feminist funds that have begun to shift their funding models towards channelling resources directly to partners in the Global South:"We've been supporting different things. One of them is just getting at the root of what we call Southern epistemologies, sort of different ways of knowing. We support an international network of Indigenous women who have what they call their intercultural sort of ways of understanding". Another donor described supporting a data equity framework, which guides research teams through equity-oriented reflection at each stage of the research process. Participants also pointed to emerging good practices from a leading humanitarian health funder, which provides seed grants to allow partners six months to collaboratively co-develop proposals. This approach was felt to foster greater ownership and create a more balanced power dynamic, potentially enhancing the ability of Southern institutions to shape research decisions. However, participants noted that such initiatives alone will not address deeper issues around resource allocation. Internally, some participants called for a reassessment of HHCC's role in global partnerships, with a member sharing: "We saw the shift in funding, the shift in power. And in some ways, I would imagine [that in] the future [...] LSHTM would still have these global links, but would act as a technical advisory role [...] that might upset lots of people. But [then perhaps] LSHTM will be much more made up of staff that reflect the global picture, [...] that we see a very different makeup of the staff at LSHTM."

## Research methods

Most participants from humanitarian contexts shared how local knowledge systems and indigenous analysis lenses are commonly overlooked, perpetuating epistemic injustices. Western research standards, with a preference for quantitative over qualitative methods, further marginalised local perspectives, according to most participants. This preference is perceived to limit reflexive processes about the power exerted by researchers in selecting which voices are amplified. One HHCC member further explained: "Understanding our voice, perspective, gaze, positionality and how that shapes the way we analyse and communicate research. And then on the other side, you know, the voice that we hear back, whose voices we centre, and how have their own individual experiences help to shape the way that they see the world? And how does that inevitably shape our kind of research findings in a very subjective way? And I think that does feed into the pursuit of a kind of decolonized science. Because it speaks to who has

power, who is controlling the narrative and so on, and who is using neglected in that process are overlooked". A senior academic HHCC partner said: "One thing that we are very aware of is [colonising data and research], where the research is happening, in these unfortunate countries where the crises are, [under] financial restraints, usually, we are used mainly as a data collectors". Participants critiqued the dominance of epidemiological and clinical training in humanitarian health, noting a limited engagement with critical equity and decolonial research methods. A HHCC member explained: "Not enough people in public health or humanitarian health have sort of studied or have knowledge of pedagogical approaches or kind of decolonial approaches to research". Moreover, the ethics approval process often overlooked decolonial approaches to research. The absence of questions and requirements related to the equitable access of partners to data, the role of partners in the research, the researcher's positionality, and returning research findings to participants were discussed as representing a missed opportunity for ethical review boards.

Despite these challenges, most HHCC members and partner organisations showed enthusiasm for using innovative and participatory research methods. A prominent example is co-production, a collaborative approach that ensures research is grounded in local priorities, expertise, and knowledge. This fosters trust and a sense of ownership within the community where research takes place. The use of anthropologically grounded methods and mixed method methodologies was also mentioned as helping to centre the experiences of communities and research participants. This approach enables a more accurate and nuanced interpretation of research findings. One external actor highlighted the value of collaborative coding as an analytical approach. This method engages local researchers in interpreting and analysing data, ensuring their perspectives and voices are central to the research process. A HHCC partner explained: "*Collaborative coding is a transformative method. Local data collectors are usually thrown around and like they're kept out of anything, they're completely alienated from the research,... but in collaborative coding, you present a local interpretation of the data and benefit from a neutral coding by LSHTM colleague as well. On top of everything, people [researchers from HHCC partner institutions] who are given the opportunity to grow research skills*".

## Management and governance

LSHTM structures and systems governing research partnerships were discussed as often disadvantaging partner institutions in humanitarian settings. Most participants referred to systematic structures governing unfair partnership contracts, data ownership/access and intellectual property rights agreements. Some participants flagged how current contracting practices are designed to favour LSHTM and are ill-equipped for fairly working with international partners. While some at HHCC have challenged these unfair clauses, such efforts come at a cost – lengthy delays in the contracting process. A few participants explained how the limited resources at many partner institutions limited their ability to effectively negotiate or challenge the British legal system and contracting systems, or suggest changes. One HHCC member explained: "*The subcontract[s] awarded by the school have a section on [...] intellectual property that awards any arising intellectual property out of a project to the School. And that's something I've stopped and complained about two times [saying that] I cannot [send such a sub-contract [...] to my partner. It is unfair*". This leaves institutions in humanitarian settings with limited space to influence the partnership dynamics, research priorities and decision-making. Another HHCC member expanded: "*For an institution with most of its work with international partners, we are ill-equipped for this, and want everyone to follow our system. And even like the fact that we don't have them [contracts] in any other languages except English, our contracting office refuses to deal with languages other than English and to try to work with partners whose operating language is not English is challenging. And then it ends up being people who speak English, likely the principal investigator from the partner institution, doing contracting. They're not the legal expert or the contract expert...this is not right*".

## Research dissemination

Some participants explained existing research norms centre research dissemination on publishing and policymaking, and less towards research dissemination among communities in humanitarian settings. One HHCC partner explained: "I think

decolonisation even goes up to the community, where you have done the research, you should be able to, to value those people, and give them feedback on what you have done". Several HHCC partners from humanitarian settings highlighted their perception of increased visibility and representation in conferences, public engagement activities, and communication with key stakeholders, including donors and funders, majorly in studies done in collaboration with HHCC. One HHCC member shared the increasing internal attention to research dissemination practices, stating "I think attempts have been made to ensure more equitable representation. And certainly [in] some of the projects that I'm thinking of. There was a recognition that findings and learnings needed to be communicated first and foremost, to the crisis affected communities". Another HHCC member described the increasing intentionality of these practices to actively challenge the historical norm of Global North institutions dominating research dissemination and dismantle perceptions of limited academic capacity in humanitarian settings.

Most HHCC partners from humanitarian settings also commended the fairness of authorship arrangements within their projects. They noted instances where researchers from their institutions were designated as first authors on publications, reflecting their fair contributions. One HHCC partner shared: "*We had very honest discussion from the beginning. We've agreed that whomever working and leading will be a co-author based on their efforts. First author doesn't have to be always from [the] London School*".

### Decolonising the curriculum, supervision and teaching

HHCC members who were engaged in teaching modules/courses on conflict and health, humanitarian health and other related areas, explained how the current model overemphasises the technical and biomedical aspects of public health while engaging less with the social and political systems that shape health. The curriculum itself was discussed as something that is steeped in a Eurocentric worldview or a foreign gaze. One HHCC member shared: "*You have to constantly be reflective on this power dynamic and what your role is within kind of public health programme design in countries worldwide.... I want the module to exist because I don't want people to go out there and make mistakes. But it's also what I am teaching students to do to maintain colonial mindsets*". Some HHCC members linked this to limited diversity among faculty and students stifles a plurality of perspectives in the classroom, in terms of ethnic, professional, disciplines, and lived experiences. One HHCC member shared: "*Not enough people have studied alternative [...] pedagogical [or decolonial] approaches [...] to education. So not only approaches to teaching, but the sort of scholarship that we need to elevate. So, it's the type of people that are doing the teaching, and their lack of access to that kind of knowledge, their lack of experience [of] decolonial pedagogies and decolonial approaches to teaching... And also, I think, you know, that they are being pressed to work within the confines [of] the way that public health is taught, which [...] has never really engaged with this question. Yes. It's always been very technical*".

A few HHCC members shared the preference to present voices from United Nations agencies or INGOs in teaching materials or guest lecturers over local voices, potentially attributed to HHCC's existing institutional relationships. Compounding this was the lack of a harmonised approach to decolonising the curriculum and teaching, and limited institutional spaces to discuss these issues. One HHCC member said: "*We may have a more diverse workforce that is still perpetuating the same injustice and the same imbalance of power and the same privilege, and we're not really having that conversation [formally] within the institution. We're having it informally. Colleagues, often behind closed doors are having these conversations, but I don't see it really being advanced by [a] kind of open formal way by the institution*".

Participants discussed how limited support from HHCC and LSHTM leadership in the form of guidance and resources left the task of curriculum review to module organisers and teaching staff. This resulted in an inconsistent approach, with some modules perpetuating exclusionary practices despite individual efforts to integrate critical content related to decolonisation and humanitarianism.

Early-career HHCC teaching staff and doctoral students/candidates shared how they received limited support and guidance from their managers/supervisors regarding teaching or curriculum. Doctoral students reported minimal discussions

with supervisors regarding how their research design, data collection, partnership formation, and consideration of research ownership by local institutions all contribute to perpetuating colonial practices. Other barriers were mentioned, such as the current employment contracts and visas hindering diversity among staff and students. The precarious employment contracts among early-career staff created a climate of fear and discouraged them from challenging power imbalances or voicing concerns about potential retaliation. One HHCC member said: "*One of the big problems is everybody's on short-term contracts [...] it's quite unstable for many people. So, to stand up for it, for example, with the strikes this week, many junior staff will be too frightened to go on strike. I think that's one of the biggest problems with the institution*". Participants said these obstacles were deeply embedded within the broader LSHTM business model, necessitating a radical change. Despite these historical and contemporary barriers, good practices emerged. Many HHCC members acknowledged the important role of student-led initiatives and informal networks that address anti-racism and decoloniality. These initiatives provided spaces for critical discussion and engagement with these essential topics. Several HHCC members mentioned the "Decolonising the Curriculum" toolkit, developed by a collaboration between the "Decolonising the Curriculum" work-stream group and the Centre for Excellence in Learning and Teaching offering guidance, links to additional resources, and examples of how staff have incorporated decolonial perspectives into their teaching practice. Similarly, the FAIR Student Toolkit "Introduction to decoloniality and antiracism in global health" was developed to facilitate student engagement with issues of race, racism, (de)coloniality, and anti-racism in the context of global health. While a few HHCC members who are module organisers reviewed the curriculum annually and incorporated more diverse literature and readings, these actions appear to be individual efforts rather than part of a broader institutional vision for decolonising the curriculum.

Finally, the presence of staff from diverse racial, ethnic and gender backgrounds was viewed as a key advantage. Some HHCC members shared that staff who are from or have prior experience working in humanitarian settings offer a deeper understanding of local actors, research capacities, priorities, languages, and policy landscapes. This deeper understanding facilitates the development of new and diverse partnerships, including collaborations with youth groups and community-based organisations. These partnerships are essential to ensure research is not conducted in isolation but directly serves the needs of the communities it aims to understand and support.

## Discussion

To our knowledge, this is the first comprehensive study that explores the institutional challenges and tensions around decolonising both research and teaching at a UK Higher Education (HE) institution, or in the field of humanitarian health. The existing literature tends to focus either on decolonising research [38,39] or teaching [40–46]. The unique aspect and strength of this paper lies in its comprehensive investigation of current and historical barriers to decolonising research and teaching, as well as other en/disabling aspects such as leadership and institutional appetite for decolonising across the entire HE institution within which HHCC is embedded in. This included considering how different areas of work are linked together, how they reproduce hierarchies, and what it means to untangle and decolonise those. Research-led teaching, for example, might reproduce potentially problematic assumptions that are embedded in our research processes in the classroom.

Our principal findings elucidated discrepancies in the conceptual understanding of decolonisation at the institutional level, and between external partners and HHCC members. Institutionally, a conflation of decolonisation with concepts like equitable partnerships, EDI, and localisation is illustrative of broader trends around decolonising terminology within the scholarship and practice of development and humanitarian assistance [29,30,47]. The conflation is driven by colonial hierarchical academic and humanitarian power structures, limited institutional appetite within the academic and humanitarian sectors, and tick-box approaches to decolonisation. While decolonisation hinges on recognising and making visible colonial assumptions and attitudes [48], approaches such as EDI or equitable partnerships tend to address isolated symptoms of the coloniality of humanitarian health, expressed in specific inequities. External partners extended the meaning of decolonising research to encompass empowering the communities where they are based and/or research is conducted.

Disagreement about the meaning of epistemological and/or material decolonisation often excluded explicit references to anti-racism [8,49,50], including to some extent at HHCC. This is symptomatic of the wider, post-Black Lives Matter literature on decolonising research, teaching, and other practices within humanitarian health, which tends to problematise racial inequalities without necessarily offering explicit, practical guidance around anti-racism [3]. Critical development scholars in contrast, have argued that the development project, of which humanitarian health is part, is a fundamentally racial project [51]. As such, development interventions continue to structure expertise, capacity and knowledge creation along racial lines [8]. In our research, however, HHCC members only identified race and/or racism as a barrier to decolonising work at the HHCC in relation to research design. This is reflective of their positionalities as researchers in a competitive HE landscape, where disrupting racialised power dynamics may produce disadvantages for individual researchers, for instance via foregoing PI positions or first authorships.

The study outputs, a charter and implementation guidance for HHCC members to decolonise their work (Box 1), provide specific guidance on how racialised power dynamics may be disrupted [32]. Racialised power dynamics characterise everyday working relationships, influence decision-making processes and produce unequal access to resources such as training, funding, remuneration, and knowledge [32].

## Box 1.  Summary of the HHCC charter and implementation guidance [32]

As a key output of this study, the HHCC Humanitarian Research Charter [32] was developed as a resource to guide ethical, contextually grounded, and decolonial approaches to humanitarian research, teaching, and partnerships—both within HHCC and its partnerships. The Charter aims to support HHCC's efforts to challenge colonial histories and legacies by embedding decolonial principles into its work. It outlines a set of concrete commitments and ways of working, intended to be regularly revisited and refined through ongoing dialogue within the HHCC community. Grounded in three core principles—decolonisation as a comprehensive practice, the disruption of racialised power dynamics, and change as a continuous process—the Charter includes commitments to be led by actors from crisis-affected contexts, to challenge the assumed neutrality of humanitarianism, to reimagine risk and capacity, and to redistribute resources more equitably. These are reinforced by ways of working that prioritise collective action among HHCC members, collaboration with other LSHTM centres and external actors, and influence across institutional, donor, and sectoral levels. The Charter is accompanied by an Implementation Guidance document that provides practical steps for operationalisation. A monitoring and learning mechanism were put in place to track progress, incorporating feedback loops, reflection sessions, and iterative revisions to ensure the Charter remains dynamic and responsive.

There is little documentation of accountability mechanisms for decolonising work across HE and the wider humanitarian sector in general to track material progress on anti-racism. Few to no studies develop such mechanisms, and we therefore invite further research into how accountability mechanisms for decolonisation efforts, including anti-racism, can be implemented effectively at UK HE institutions and beyond. More work needs to be done to consider how accountability mechanisms can avoid replicating tick-box approaches or use tools that are grounded in coloniality (such as log frames) to measure change.

The market-driven academic landscape acts as a significant structural barrier to decolonising research and scholarship about humanitarian settings. LSHTM has extensive links and institutional partnerships with crises-affected countries; however, participants argued that promotion frameworks, publication standards, and grant conditions favour first authorship and PI experience. Consequently, these institutions rarely lead collaborative research efforts at LSHTM or HHCC. Funding structures exacerbate these inequities, as high-level funders shape what funding can be offered by whom. Such inequitable North-South partnerships perpetuate colonial understandings of expertise and capacity as Global North institutions,

researchers and authors continue to lead humanitarian health interventions in crises-affected communities and countries in the Global South.

Our participants highlighted how short funding call deadlines in particular present structural barriers to involving communities in countries affected by humanitarian crises in early stages of the research process, particularly in problem identification and defining research questions. This issue raises important questions around who is setting the research agenda and produces knowledge in the first place [17]. Consequently, rather than coming out of needs and experiences of crises-affected communities or countries, global/humanitarian health scholarship is predominantly produced by Global North actors [7]. However, external partners and internal members at HHCC have highlighted research partnership practices at the centre that work towards addressing this epistemic justice issue and include the voices of the research participants and partners from the early stages of research design. These good practices include using project partner-led steering groups for research governance, shared leadership and decision-making, and naming co-PIs. To revert the foreign gaze in humanitarian health research and ensure that research is led by those from crises affected countries, we recommend the meaningful participation of crises-affected communities and partners across all stages of the research (problem identification, defining the research question, design, fund acquisition, data collection, analysis and dissemination), as stated in the HHCC Charter and implementation plan [32]. Ultimately, this would produce needs-based research outputs and challenge epistemic injustice in humanitarian health scholarship. The study participants furthermore highlighted the need to reform institutional ethics approval processes to include requirements around equitable partnerships and returning research findings to participants as another practical way to contribute towards decolonising humanitarian health research.

As suggested by participants of this research, these efforts need to be complemented by institutional leadership advocacy towards donors and funding bodies to reform policies and practices that perpetuate colonial partnership models. Our findings are reflective of other work that identifies the requirement of UK-based institutions to be lead applicants on major research projects, as well as funders' relatively short timeframes for developing research proposals as part of the reason why Global North perspectives continue to be prioritised over needs-based research led by the Global South [52]. Various practical suggestions exist for donors to localise Calls for Papers and review processes, for example by including local civil society organisations in the design of both [52], or by inviting Global South researchers to funder advisory panels [17]. Stakeholder feedback from our research further demonstrates how researchers can be responsive to local needs within the restraints of the current funding landscape, for instance by collaboratively planning with crises-affected communities during study inception and research design, or using innovative and participatory approaches such as co-production.

External partners commended co-planning and co-production methods employed by HHCC researchers as particularly responsive to community needs. These methods involve stakeholders in all stages of the research process, from the initial design stages onwards, and amongst others, include setting up strong governance structures, such as steering groups. Clearly defining roles and responsibilities, collectivising decision-making and sharing project ownership are an important practical step towards decolonising research, identified as such in the relevant literature [3,15,37,39,53]. In our example, it has clearly promoted a sense of shared ownership between HHCC and external collaborators. Furthermore, and according to partners from humanitarian settings, HHCC members engage in equitable publication, public engagement and dissemination practices, including in conversations with donors. These practices actively challenge the historical norm of Northern institutions dominating the dissemination of research findings and dismantle paternalistic perceptions of limited academic capacity in humanitarian research settings.

At HHCC, we have thereby identified examples of designing and conducting needs-based, country-responsive and participatory research that is led by communities from crises-affected contexts. However, these individual efforts strongly contrast with a lack of appetite at the LSHTM leadership level to engage with decolonising humanitarian health research and teaching. What are the consequences of that lack of leadership on implementing far-reaching, sectoral change towards decolonising humanitarian health? The results of this study suggest that without clear institutional leadership, which includes channelling funding into decolonising efforts, such exemplary efforts will remain fragmented and isolated.

They may even result in career disadvantages for researchers who engage in equitable publication practices who as a result may forego academic prestige required for obtaining further research funding and for meeting promotion requirements [54,55].

Furthermore, there are serious concerns, expressed by our participants but also discussed in the wider literature on decolonising global health [56,57] around the way in which HHCC as a Western institution takes this kind of leadership, even on an ad hoc basis, on decolonisation efforts. We recommend that future research pays particular attention to where the impetus for these efforts is coming from, and whether those initiatives are taking away conceptual or empirical space from initiatives led by racialised and/or otherwise marginalised researchers and communities to develop their own initiatives, especially in the Global South. The external partners in our study suggested that their limited knowledge of decolonising initiatives in Africa in particular may be due to more pressing concerns around low visibility of local issues and concerns. Southern researchers play an active role in developing Global South-led decolonising initiatives, for example, including the degree to which these efforts can be achieved independently of whether Global North researchers are decolonising their work. Study participants did not mention the influence of powerful financial and corporate interests that are accumulating and extracting wealth from the global/humanitarian health sector [58] as a constraining factor for Global South-led decolonisation efforts, but this may need further consideration.

However, despite these structural barriers, this study has also identified opportunities that Global North institutions like LSHTM can capitalise on given their significant power and leverage in the global field of public and humanitarian health, including with donor agencies. This includes advocating the core principles identified in this research and laid out in the HHCC Charter [32] to LSHTM's internal and external stakeholders to influence policy and practices towards equitable and decolonial futures. For example, our research recommends LSHTM use its powerful position to advocate for donor reforms to policies and requirements that channel funding towards high-income institutions, such as what kinds of actors can hold funds, who must be PI/CO-I, information sharing restrictions, short funding call timelines and inflexible project management templates. In terms of internal reforms, simplifying contract, accountability, cost recovery and data ownership policies as well as reformed due diligence and accountability processes would contribute towards moving away from a racist and paternalistic model that institutionalises research and collaboration with non-Western partners as inherently risky.

Finally, colonial legacies are also reflected in the teaching provision, staff diversity, supervision and curriculum design at HHCC. The prevalence of teaching technical and biomedical aspects of humanitarian health can be said to be a legacy of a Eurocentric, 'White Saviourism' approach to humanitarian aid and assistance [4,59]. In teaching, the racist and paternalistic trope of portraying communities of colour in conflict-affected settings as 'beneficiaries' removes their agency and power while a benevolent understanding of 'aid' sidelines histories of colonial exploitation [3]. Teaching critical social science approaches to humanitarian health would challenge the assumed neutrality of humanitarian practice, which protects the status quo of colonial and racial(ising) hierarchies between those who give and those who receive humanitarian assistance [32,55]. Dismantling these colonial legacies in teaching provision would require educators to include knowledge sources that incorporate context-specific and regional expertise to balance or challenge dominant Western knowledge production, and to review reading lists for dominant and excluded voices and narratives. An important finding we identified was appetite by doctoral students to provide doctoral research supervisors with relevant training or guidelines to take a decolonial approach in supervising all aspects of the doctoral research process (research design, methods, dissemination, etc.). These recommendations are currently being implemented within the 'Decolonising Doctoral Supervision' project at LSHTM.

At HHCC, particular weight is given to INGO voices in teaching materials or guest lectures, as opposed to local knowledge systems or community voices, which we argue is a result of the research-informed nature of teaching in UK higher education. If, as discussed above, research is not led by affected communities, grassroot organisations and/or national institutions, then this is likely reflected in teaching practices. At LSHTM, various student and staff-led initiatives have

provided decolonial teaching and student toolkits; however, in the absence of institutional guidance and executive leadership on how to mainstream these approaches, efforts to implement those will remain isolated. Practically, more holistic efforts are needed to decolonise teaching provision, including, amongst others, decolonial classroom management training (e.g., inviting student feedback, increasing student participation and confidence, disrupting dominant voices etc.), recruitment practices that create a supportive environment for diverse applicants, and ensuring a robust formal complaints procedure to address racism, and/or misogyny within the teaching team and/or environment.

Based on our findings, the study offers a set of recommendations to guide decolonising efforts in humanitarian health research, teaching, and partnerships. These recommendations aim to support HHCC, LSHTM, and beyond - in challenging colonial power structures and embedding decolonial, that is, equitable and contextually-grounded practices into their work. These recommendations are summarised in Table 2.

Through a reflexive process, we critically engage with the politics and power dynamics that underpin the research process and this partnership model. We reflect on the inherently political nature of conducting such work, recognising the multiple layers of politics and tensions within an institution like LSHTM. These tensions manifest at various levels: within the hierarchical structures of the research team; in the dynamics with HHCC, a centre led by two of the study's co-authors, who also served as the PI and co-authors of this study; and, crucially, between FAIR—an independent student-staff network committed to anti-racism and the decolonisation of global and humanitarian health—and LSHTM, whose financial and reputational priorities can at times conflict with the goals of the decolonisation agenda.

This study has several limitations. The first is the context-specific nature of the study and its narrow focus. We engaged researchers and partners from a specific, relatively small research centre that is part of a much larger, research-intensive Higher Education institution in the UK, making it challenging to disentangle centre practices from the broader institution. Our findings may be limited in their applicability to other contexts, though we hope to make important contributions to decolonising humanitarian health broadly. Furthermore, the study did not include any student perspectives beyond doctoral research degree students. LSHTM only has postgraduate students, though a future research project may also benefit from engaging the broader student body in the endeavour of understanding how research and teaching of humanitarian health can be decolonised. Another limitation is that the study did not include race as a specific analytical category, contrary to recommendations around decolonising humanitarian health [3,15]. This was because our intention was to focus on institutional structures and barriers to decolonising humanitarian health within HHCC. Our primary category of analysis was therefore whether participants were internal or external, though we were able report on what participants said about race. Lastly, our study is limited by institutional power hierarchies, including how LSHTM as an institution is perceived and positioned. We suggest these hierarchies may have affected how participants, including external stakeholders, engaged

**Table 2. Key recommendations.**

- Decolonisation should be framed explicitly as a transformative process rooted in power and justice and avoid framing tick-box approaches as solutions to coloniality.
- Institutional leadership should actively advocate towards donors and funding bodies to reform institutional policies and practices that perpetuate colonial partnership models.
- Ensure the meaningful participation of crises-affected communities and partners across the different research stages, in order to try to minimise the foreign gaze in humanitarian health research and support leadership by those from crises-affected countries.
- Advocate to donors and funders for changes in funding eligibility, contracting, and due diligence requirements to enable equitable partnerships, and restructure incentives and policies within universities and donors to dismantle racialised hierarchies.
- LSHTM should use its powerful position to advocate for donor reforms to policies and requirements that channel funding towards high-income institutions, such as what kind of actors can hold funds, who must be PI/CO-I, information sharing restrictions, short funding call timelines and inflexible project management templates.
- Integrate critical, regionally grounded knowledge and pedagogies into humanitarian health education, ensuring that curricula reflect diverse perspectives and disrupts dominant narratives.
- Further research should be conducted into how accountability mechanisms for decolonisation efforts, including anti-racism, can be effectively implemented at UK HE institutions and beyond.

with the topic and the extent to which they felt comfortable to share their experiences. Additionally, unlike institutions that externally commission work on anti-racism, decoloniality, and equity, diversity, and inclusion (EDI), the collaboration between HHCC and FAIR underscores the opportunities and challenges of driving meaningful change from within and challenging institutional norms. While such collaborations hold the potential to amplify marginalised voices, we suggest that ultimately efforts to bring about meaningful change may be limited without real institutional backing – which has implications for other studies and other centres at LSHTM and similar institutions who may want to commission similar work. Conducting studies like this is inherently political, due to the multiple layers of politics and tensions within an institution like LSHTM. These dynamics meant that HHCC and FAIR were continuously negotiating their power and positionality as two entities with different mandates, but working together towards a shared goal.

## Conclusion

To our knowledge, this is the first comprehensive empirical study into decolonising research and teaching within a UK HE institution, or within a humanitarian centre. Our study contributes new insights on how the intersection of research, teaching, and the wider humanitarian donor and practitioner landscape produces structural barriers to dismantling colonial hierarchies. We draw attention to the market-driven academic landscape and funding structures that inevitably shape who sets research agendas, who produces knowledge, who is viewed as having expertise and authority to teach in academic spaces, and the extent to which crises-affected populations can meaningfully be involved in research and teaching on humanitarian health. While recognising the importance of individual-level initiatives to advocate for change, to challenge traditional research methods and to incorporate practical strategies to challenge colonial and racialised power hierarchies, the extent to which power hierarchies can be challenged depends significantly on institutional appetite. The risk for institutions like LSHTM is that efforts to decolonise may remain fragmented or isolated to the work of individuals. There is a need for a clear agenda and accountabilities for decolonising research and teaching, backed by institutional leadership. LSHTM holds a powerful position within public health spaces and has significant potential to play an active role in advocacy – especially in funding spaces - that remains largely untapped. Political will is critical to unravelling institutional legacies of colonialism, and any institution committed to decolonial practice must recognise and seek to tackle this.

## Author contributions

**Conceptualization:** Amber Clarke, Katharina Richter, Michelle Lokot, Althea-Maria Rivas, Neha S. Singh.

**Data curation:** Sali Hafez.

**Formal analysis:** Sali Hafez.

**Funding acquisition:** Amber Clarke, Neha S. Singh.

**Investigation:** Sali Hafez.

**Methodology:** Sali Hafez, Katharina Richter, Michelle Lokot, Neha S. Singh.

**Project administration:** Amber Clarke, Katharina Richter.

**Resources:** Amber Clarke.

**Supervision:** Amber Clarke, Michelle Lokot, Neha S. Singh.

**Validation:** Katharina Richter, Michelle Lokot, Althea-Maria Rivas, Neha S. Singh.

**Writing – original draft:** Sali Hafez, Amber Clarke, Katharina Richter.

**Writing – review & editing:** Sali Hafez, Amber Clarke, Katharina Richter, Michelle Lokot, Althea-Maria Rivas, Neha S. Singh.

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
