## [Decision Letter · Decision Letter 0]

PGPH-D-25-00256

Dismantling colonial legacies: Decolonising research and teaching at the Health in Humanitarian Crises Centre, London School of Hygiene and Tropical Medicine

Dear Dr. Hafez,

Thank you for submitting your manuscript to PLOS Global Public Health. After careful consideration, we feel that it has merit but does not fully meet PLOS Global Public Health’s publication criteria as it currently stands. Therefore, we invite you to submit a revised version of the manuscript that addresses the points raised during the review process.

Please address all the comments by both reviewers carefully. Have clear definitions with citations of key terms used (e.g., global health, humanitarian health). Strengthen the clarity and details in the methods section as per reviewer comments.

We look forward to receiving your revised manuscript.

Kind regards,

Shashika Bandara

Academic Editor

Journal Requirements:

Additional Editor Comments (if provided):

Reviewers' comments:

Reviewer's Responses to Questions

**Comments to the Author**

1. Does this manuscript meet PLOS Global Public Health’s publication criteria ? Is the manuscript technically sound, and do the data support the conclusions? The manuscript must describe methodologically and ethically rigorous research with conclusions that are appropriately drawn based on the data presented.

Reviewer #1: Yes

Reviewer #2: Yes

2. Has the statistical analysis been performed appropriately and rigorously?

Reviewer #1: N/A

Reviewer #2: N/A

3. Have the authors made all data underlying the findings in their manuscript fully available (please refer to the Data Availability Statement at the start of the manuscript PDF file)?

Reviewer #1: No

Reviewer #2: Yes

4. Is the manuscript presented in an intelligible fashion and written in standard English?

Reviewer #1: Yes

Reviewer #2: Yes

5. Review Comments to the Author

Reviewer #1: Summary

Thank you for the opportunity to review this article. This research is an important contribution to how Global North institutions have been and should be conceptualizing and enacting decolonization in the fields of global and humanitarian health research. The findings from this research will be helpful to inform future action as to how institutions can support the decolonization agenda in humanitarian health research and how they can address limitations in doing so.

Abstract

- Clearly specify who is being referred to within the humanitarian health community—whether the discussion includes researchers, academics, practitioners, policymakers, or other groups.

- Clarify the context in which the decolonization discourse is being discussed. At times, it is unclear whether the authors are referring to decolonization in academic spaces, fieldwork, or both. Providing explicit distinctions will enhance the reader’s understanding.

Introduction

-The authors appear to use "global health" and "humanitarian health" interchangeably. However, these sectors have distinct origins and mandates, and the decolonization discourse has evolved differently in each. In the first paragraph, the authors introduce decolonization in global health while referencing both sectors, but later shift focus exclusively to global health. To improve clarity, the authors could explicitly acknowledge and differentiate between both sectors, ensuring clarity when discussing the decolonization discourse in each. Currently, the text moves between them without clear distinctions.

- When discussing white saviorism, it would be valuable to clarify that this concept is not solely tied to white identity. Rather, it refers to a broader mindset of individuals and groups who hold power thinking are "saving" those in vulnerable positions—one that can be perpetuated by anyone in a position of power, regardless of race or ethnicity. Acknowledging this nuance would help expand accountability, including for local elites who may also reinforce white savior dynamics in humanitarian health contexts.

- Localization - could you briefly explain more how localization is claimed to perpetuate the existing power dynamics (p.7)

Methods

- Could the authors specify whether the results of the scoping review also integrated within this article, or how they have informed and influenced the development of this paper? In other words, please explain the role the scoping review had prior or during the development of this qualitative study.

- Could the authors clarify whether the term "donors" refers to those funding academic institutions, humanitarian organizations, or both? A brief explanation would help distinguish the different roles and influences of donors in these spaces

- To avoid confusion on table 1, I may suggest to change the wording to mixed race and add in parenthesis (Asian and White)

- Could the authors add some reference to their participatory research approach and add more details as to what this entailed throughout the research process?

- Could the authors provide more details on the 'Data Collection and Analysis' section in the Methods. Particularly, did the authors use any methodology to conduct the qualitative study? How did the inductive coding lead to the selected themes, and was this conducted by 1 or 2 researchers? What actions did it involve to code inductively? What frameworks or research questions were used to conduct the deductive coding?

Results

- Would it be possible to add a quotation or example of this claim on pp. 12: “Discussions within HHCC and LSHTM participants often conflated decolonisation with concepts like equitable partnerships, Equity, Diversity, and Inclusion (EDI), and localisation.” It is an important point that is referenced throughout the article, but little to no quotations are offered to support it.

- In the ‘Funding’ section, starting on page 18, is there any perspective from the donors interviewed worth sharing? It would be valuable to incorporate the perspective of their own role in funding structures and decolonization.

Discussion

- The authors provide several recommendations and suggestions throughout the discussion. Would they consider synthesizing these in a table? This could help highlight key takeaways and provide a clearer guide for those engaging in humanitarian health research and decolonization.

References

- Please ensure the DOI and/or volume and page numbers are available for all Journal Article references

Reviewer #2: Overall, this is a very well written, coherent, and meaningful piece of research. I recommend accepting this manuscript for publication, and I have only minor comments and suggestions for the authors to consider. I look forward to seeing this published and citing it in future work. Kudos to the authors for their high level of commitment and integrity, both to decolonising these fields and to the research process - the data speaks for itself.

Introduction: All of the important concepts are laid out in this section. However, this study focuses on “the humanitarian health sector specifically” (p.8). I am thus wondering if you need to refer to global health (or DGH) at all in your introduction, or if it makes more sense conceptually to dive straight into the sub-section on ‘Colonial Legacies in the Humanitarian Sector’ (and bring in the paragraph starting "Aligning with Aloudat's..." here instead), then ‘Decolonisation Efforts at LSHTM’, then ‘Gaps and Divergent Conceptions…’. I defer to the authors expertise and overarching views of this topic(s) and field(s). Otherwise, for readers like myself who work predominantly in global health and not humanitarian health, perhaps providing a conceptual link between global and humanitarian health (eg, similar and different colonial impacts, legacies, patterns, etc) in the introduction would suffice.

Methods: Only minor feedback. Overall, this section is written clearly.

1. When explaining the Study Design (p.9), I noticed there is no information on participant selection. Would you consider elaborating how participants for the interviews and FGDs were selected ie, were external stakeholders informed by the scoping review? Was the selection purposive? Was maximum variation sampling used? Or snowballing? Were there dependencies between the research team and participants? Were interview participants different or the same as the FGD participants? One sentence on the approach would help to clarify.

2. In Table 1 (p.9), the formatting makes it look as if 'White' is duplicated and there’s no data for ‘Mixed Asian’. I’m also curious as to whether participants had the opportunity to self-define their racial / ethnic backgrounds, or were these categories assigned by the research team? A footnote to the Table might be helpful here.

3. When explaining the FGD methodology (p.10), only the first and second FGDs are described. Was the participatory charter design workshop considered the third FGD? Clarification of the third FGD would help to finalize this paragraph.

4. When explaining Data Collection & Analysis (p.10), I’d recommend using consistent terminology and not confusing codes with themes ie, “This process involved the development of a codebook informed by the literature review, with additional key CODES…”. I would also avoid saying “we intended to extract race-related THEMES” as themes are not strictly extracted from data, but interpreted or synthesised from the codes. You might consider replacing ‘extract' with 'develop’. As a side-note, is there a need for this sentence? It reads a bit odd that you would mention which themes you did NOT develop, but not state which themes you DID develop — for the authors consideration.

5. The positionality paragraph (p.11) is well received, I appreciate the authors thoughtfulness and diligence in making these statements.

Results: Brilliant results - very meaningful, and communicated with great clarity and transparency. Only minor feedback.

1. I think the second and third themes are well named in that they summarize the main content of the subsequent paragraphs. But the first theme is a bit ambiguous ie, ‘Definitions and institutional readiness’ - of what? Would the authors consider a more specific phrasing for the first theme, something like ‘Defining, understanding, and driving decolonisation efforts institutionally’ (just an example). I think the first theme would also benefit from sub-headings, as done in the second theme.

2. I’m curious about how the authors decided to attribute some quotes to interviewees ‘of colour’ or ‘white’, and not others. For example, the quote at the top of p.13 does not state the racial / ethnic background of the interviewee whereas many on p.12 do. I am not recommending the authors commit to one or the other, rather find some way of being consistent, perhaps where or if there are sharp contrasts. (My own opinion, if helpful, is that it is interesting to declare for all quotes. This could be done using participant codes after each quote.).

3. The paragraph starting “Partners…” (p.15) has language that would be more fitting in the Discussion ie, “This optimism LIKELY…” rather than “This optimism stems from…”. Same later on, “These positive experiences MAY HAVE…”, and “INTERESTINGLY, some HHCC partners…”. Would suggest removing words that imply the authors’ own interpretations, as I think the data supports these statements.

4. Noticed grammar and punctuation potentially missing from some quotes ie, “The historical [insert comma] the colonial history of LSHTM…” (p.16) and “… who have decided. [Capitalise] that we are the…” (p.17). Perhaps check all quotes.

5. The sentence starting “One HCC member shared the increasing internal attention…” (p.23), do you have a supporting quote for this? There’s a lot said and by just one interviewee, I think a quote would help illustrate.

Discussion: Every point made in the discussion is valuable. However, currently, the discussion flows from conceptualisation to accountability to funding structures to leadership and advocacy to co-planning/-production to partnership dynamics to opportunities to teaching, staff diversity, supervision, and curriculum. The Charter is introduced at different points, and recommendations are woven in here and there. Would the authors consider looking at the structure of the discussion? Some suggestions below.

1. As a key study output, the Charter is very interesting and sounds like a great resource. Could this be introduced earlier in the discussion with a panel or box that contains more details ie, its purpose or mandate, goal or aims, objectives, implementation, monitoring? Text on the Charter (p.28, 29, 32) could be removed from the narrative and inserted into this panel / box.

2. Could the authors also consider a separate panel or box for key recommendations from this study, or ‘good practices’ as mentioned in the Abstract? I noticed that recommendations are interspersed (p. 28, 29, 30, 31, 32, 33) when they could be consolidated in a panel / box, perhaps merged with the Charter. Similarly then, text on recommendations could be removed from the narrative.

3. The authors mention “all stages of the research process” (p.29). What are these stages, and do the main barriers articulated in the discussion map to these stages? This might be one way to bring a bit more structure to this section, starting with the first stage in this process.

4. The paragraph starting “Unlike institutions…” (p.34) is partially repeated halfway down the next page.

6. PLOS authors have the option to publish the peer review history of their article (what does this mean? ). If published, this will include your full peer review and any attached files.

**Do you want your identity to be public for this peer review?** For information about this choice, including consent withdrawal, please see our Privacy Policy .

Reviewer #1: **Yes: ** Isabel Munoz Beaulieu

Reviewer #2: No

---

## [Editor Report · Decision Letter 1]

Dismantling colonial legacies: Decolonising research and teaching at the Health in Humanitarian Crises Centre, London School of Hygiene and Tropical Medicine

PGPH-D-25-00256R1

Dear Ms Hafez,

We are pleased to inform you that your manuscript 'Dismantling colonial legacies: Decolonising research and teaching at the Health in Humanitarian Crises Centre, London School of Hygiene and Tropical Medicine' has been provisionally accepted for publication in PLOS Global Public Health.

Best regards,

Shashika Bandara

Academic Editor

Great work in addressing reviewer and editor comments. This will be an important article in how we approach our work in global health. Thank you for all your efforts to push the field forward.